# TwoStepCisMR: A Novel Method and R Package for Attenuating Bias in *cis*-Mendelian Randomization Analyses

**DOI:** 10.3390/genes13091541

**Published:** 2022-08-26

**Authors:** Benjamin Woolf, Loukas Zagkos, Dipender Gill

**Affiliations:** 1School of Psychological Science, University of Bristol, Bristol BS8 1TH, UK; 2MRC Integrative Epidemiology Unit, University of Bristol, Bristol BS8 1TH, UK; 3Faculty of Epidemiology and Population Health, London School of Hygiene and Tropical Medicine, London WC1E 7HT, UK; 4Department of Epidemiology and Biostatistics, School of Public Health, Imperial College London, London SW7 2AZ, UK; 5Chief Scientific Advisor Office, Research and Early Development, Novo Nordisk, 1050 Copenhagen, Denmark

**Keywords:** Mendelian randomisation, drug-target validation, sensitivity analyses

## Abstract

Mendelian randomisation (MR) is an increasingly popular method for strengthening causal inference in epidemiological studies. *cis*-MR in particular uses genetic variants in the gene region of a drug target protein as an instrumental variable to provide quasi-experimental evidence for on-target drug effects. A limitation of this framework is when the genetic variant is correlated to another variant that also effects the outcome of interest (confounding through linkage disequilibrium). Methods for correcting this bias, such as multivariable MR, struggle in a *cis* setting because of the high correlation among genetic variants. Here, through simulation experiments and an applied example considering the effect of interleukin 6 receptor signaling on coronary artery disease risk, we present an alternative method for attenuating bias that does not suffer from this problem. As our method uses both MR and the product and difference method for mediation analysis, our proposal inherits all assumptions of these methods. We have additionally developed an R package, TwoStepCisMR, to facilitate the implementation of the method.

## 1. Introduction

Mendelian randomisation (MR) is an increasingly popular genetic epidemiological method used to infer the causal effect of a modifiable exposure on an outcome of interest. MR uses genetic variants as instruments in an instrumental variables (IV) analyses [1,2,3]. An IV analysis will provide valid causal effect estimates when (1) the instrument is robustly associated with the exposure of interest, (2) there are no instrument-outcome confounders, and (3) the instrument can cause the outcome only via the exposure. 

An MR analysis is ‘*cis*’ when genetic variants close to a target gene region are selected as instruments. A major source of bias in *cis*-MR is confounding by a genetic variant in linkage disequilibrium (LD, Figure 1) [4]. Although it is possible to explore this using colocalization, this approach is not able to generate MR estimates that account for any such bias [5]. Multivariable Mendelian randomisation (MVMR) is an extension of MR to account for multiple exposures. MVMR is often used for mediation analysis, but has more recently also been widely used to attenuate potential biases related to pleiotropic associations of genetic variants [6]. However, its application to a *cis* setting is limited by the requirements that there are multiple variants and that the variant-phenotype associations for the independent variables are not collinear. As genetic variants within a *cis* region are often correlated, it is difficult to find sets of variants which strongly associate with both exposures of interest and do not have collinear effect estimates.

An alternative to MVMR for performing mediation analyses is two-step MR (TSMR) [7]. TSMR leverages the product and difference of coefficients methods to perform mediation analyses using MR estimates. TSMR has not, to our knowledge, been used to attenuate bias, but we believe that the product of coefficients method could be used for this purpose in *cis*-MR, which we name Two-Step *cis*-MR (TSCMR). Specifically, the crude variant-outcome association can be decomposed into (1) a path from the variant to the outcome via the exposure, and (2) a path from the variant to the outcome via the confounder (Figure 1 and Figure 2). It follows that if we can estimate the second path, we could derive an unbiased variant-outcome association estimate by subtracting the effect through the second path from the total effect of the variant on the outcome. This adjusted variant-outcome association could then be used in a *cis*-MR analysis. Intuitively, because we would have removed the effect of any pathway that goes through the confounder phenotype, the revised MR estimate should be unbiased. The second path is itself the product of the effect of the variant on the confounder and the effect of the confounder on the outcome. The first of these will be directly estimated by a genome-wide association study (GWAS) of the confounder. The second could be estimated through MR of the confounder phenotype on the outcome. The standard error for the variant-outcome association should then be updated using either the propagation of error method or via bootstrapping (see the Appendix A for more details). 

Here, we demonstrate this novel approach, TSCMR, through simulation experiments and an applied example considering the effect of interleukin 6 receptor signaling on coronary artery disease risk. We have additionally developed an R package, TwoStepCisMR, to facilitate the implementation of the method.

## 2. Materials and Methods 

### 2.1. Methods Overview

A formal proof of the validity of using two-step methods for mediation analyses when using an MR framework can be found elsewhere [8]. We therefore used a simulation and an applied example to explore the use of TSCMR further. In brief, our primary simulation modelled a situation where a second phenotype is either pleiotropic or caused by a variant in perfect LD with the causal variant for the exposure of interest (represented by the Directed Acyclic Graph, DAG, in Figure 2). We estimated the mean bias and precision of TSCMR compared to traditional *cis*-MR after 100,000 repetitions (Further details provided below). In the Appendix A we describe an additional simulation in which we explore the application of TSCMR to settings where there are multiple biasing pathways by using TSCMR to sequentially adjust variant-outcome associations for each proposed biasing pathway (see Appendix A for visual representations).

As an applied example, we then looked at the effect of interleukin 6 receptor (IL6R) signaling on coronary artery disease (CAD) risk, and whether the genetic association was confounded by a variant in LD that associated with type-2 diabetes (T2D) risk. As T2D can also cause CAD, there is potential for confounding by LD. Reporting based on the STROBE-MR guidelines are provided below [9].

### 2.2. Primary Simulation Methods

We report our simulations using the ADEMP (aims, data-generating mechanisms, estimands, methods, and performance measures) approach [10]:

Aims: The aim of this simulation was, as a proof of concept, to explore if Two-step *cis*-MR can be used to attenuate bias in a *cis*-MR analysis. 

#### 2.2.1. Data-Generating Mechanisms

The directed acyclic graph (DAG) in Figure 2 was used as the basis for the data generation mechanism for the simulation. The nodes were simulated as standard normal variables (mean = 0, standard deviation [SD] = 1). Each arrow was used to represent a linear causal effect of the node from which the arrow originated on the node to which the arrow was directed, with each causal effect value being set as 1. The exception to this was the *cis*-variant, which was simulated as a three-level categorical variable with a minor allele frequency of 0.5 (standard deviation = 0.1), and a mean effect size of 0.1 (standard deviation = 0.05). The variant-exposure association, variant-outcome association(s), variant-confounder association(s), were each estimated in independent samples of 200,000 participants. The simulations were repeated 100,000 times. 

#### 2.2.2. Estimands and Other Targets

The estimand was the average causal effect of intervening and changing the exposure of each individual from its observed level x by a single unit, or E[Y(X = x)] − E[Y(X = x − 1)] using potential outcomes framework. 

#### 2.2.3. Methods

We compare two methods of estimating the variant-outcome association: (1) the crude association that would be derived from a GWAS, (2) the adjusted association that would be derived by using Two-step *cis*-MR. The confounder-outcome association was estimated as the Wald ratio of the association of the genome-wide genetic liability for the confounder with the outcome divided by the genetic liability-confounder association. 

#### 2.2.4. Performance Measure

The performance measure was the estimate of the causal effect and its standard error for the crude and Two-step *cis*-MR methods.

### 2.3. Applied Example Methods

#### 2.3.1. Overview

In our applied example, we sought to explore the extent to which type 2 diabetes liability confounds the association between IL6R signaling and CAD. We first ran a traditional *cis*-MR of the effect of IL6R signaling (proxied using C-reactive protein [CRP] levels) on CAD risk. We then used Two-step *cis*-MR to adjust the variant-CAD associations for any bias through type 2 diabetes liability. We used variant-type 2 diabetes information extracted from a GWAS, and an MR analysis of the effect of type 2 diabetes liability on CAD risk as the sources of additional information needed to run Two-step *cis*-MR.

#### 2.3.2. Data Sources

We used serum C-reactive protein levels as a proxy for IL6R signaling. We extracted CRP GWAS data (OpenGWAS ID: ukb-d-30710_raw) from the Ben Neale lab round 2 analysis [11]. This GWAS included around 469,000 males and females of European ancestry. Samples were measured at recruitment. Effect estimates are on the mg/L scale. 

We used van der Harst and Verweij (2017) GWAS as a source of genotype-CAD associations (OpenGWAS ID: ebi-a-GCST005195) [12]. This GWAS was a meta-analysis of the CARDIoGRAMplusC4D and UK Biobank GWASs. Both of these studies are European samples of men and women. This meta-analysis included 122,733 cases and 424,528 controls. The effect estimates were on the logOR scale. More details can be found in the original manuscript [12].

Information on variant-(type-2-)diabetes associations were extracted from the FinnGen round 5 GWAS of diabetes (OpenGWAS ID: finn-b-T2D) [13]. This GWAS has information on 29,193 cases and 182,573 controls, drawn from the Finnish general population. Cases are inferred from medical records. The effect estimates were on the logOR scale. 

#### 2.3.3. Instrument Construction

We selected genetic instruments which had a genome-wide significant (*p* < 5 × 10^−8^) association with the exposure (i.e., CRP or type 2 diabetes). We additionally clumped the variants using an r^2^ of 0.01 and KB of 10,000. We used the TwoSampleMR R package to harmonise the two GWASs and to infer which strand was positive. SNPs which were associated with type 2 diabetes but were missing in the CAD GWAS were imputed for the analysis of the effect of diabetes on CAD using an r^2^ of 0.8 from the European subsample of the 1000 genomes project. 

#### 2.3.4. Statistical Methods

The primary MR estimator was the Wald ratio, defined as the variant-outcome association divided by the variant-exposure association. We repeated the analysis calculating the Wald ratio, dividing the adjusted variant-outcome association, obtained conducting TSCMR, by the variant-exposure association.

We used four methods of meta-analysing the SNP specific Wald ratios for the effect of type 2 diabetes liability on CAD and IL6R signaling on CAD: inverse variance weighed (IVW), MR-Egger, weighted median, weighted mode. Unlike IVW, the latter three methods can return the true effect if some of the instruments are invalid, but have reduced power. Weighted mode assumes that the modal effect size is a valid estimate of the true effect size, while weighted median assumes that at least half of the SNPs are valid. MR-Egger is unbiased as long as the variant-exposure effect size is independent of the size of any bias (such as a pleiotropic effect) [14]. When the Cochrane Q statistic indicated that there was pleiotropy, we used the weighted median as the primary estimator. 

#### 2.3.5. Assessment of Assumptions

Weak instrument bias is inversely proportional to the F-statistic of the variant-exposure association, which we estimated as the square of the variant-exposure association estimate divided by the square of the standard error of this association. We also used the Cochrane’s Q statistic of the Wald ratios as a falsification test for horizontal pleiotropy in the SNP-outcome association. 

## 3. Results

### 3.1. Simulations

In the primary simulation, the crude *cis*-MR estimate was highly biased, with an average inflation 38% when compared to the true effect (mean estimate = 1.377, mean 95% CI 1.099 to 1.655, true effect = 1). TSCMR completely attenuated the bias in the crude estimate with only a moderate reduction in precision (mean estimate = 1.000, mean 95% CI 0.657 to 1.343, Table 1). We therefore believe this result supports the use of TSCMR to attenuate bias in settings where there is a single known biasing pathway.

In our additional simulation we explored the application of 2SCMR to settings in which there were multiple potential biasing pathways. In the setting in which there were two independent pathways, TSCMR slightly over adjusted, although by an amount which was far less than the standard error in either the unadjusted or adjusted estimate (mean unadjusted estimate = 2.336, mean standard error (SE) of unadjusted estimate = 0.174, mean adjusted estimate = 0.982, mean SE of adjusted estimate = 0.226, true effect = 1). On the other hand, when the two pathways were not independent, there was a large increase in the amount of overadjustment in the 2SMCR estimate (mean adjusted estimate = 0.635, Table 2). This implies that applications of 2SCMR to settings where there is more than one biasing pathway must ensure that each pathway is independent of each other.

### 3.2. Applied Example

We found evidence of heterogeneity in the SNP specific MR estimates for the Type-2 Diabetes-CAD association (*p* < 0.0001). We therefore used the Weighted Median estimate for the second step of the 2SCMR adjustment, although all estimators provided consistent estimates. Since any biasing effect between IL6R signaling and T2D should be locus specific, the use of pleiotropy robust methods should have the additional advantage of providing robustness to this bias without excluding *cis*-specific SNPs from the genome-wide analysis. 

Our unadjusted analysis found that every mg/L increase in CRP due to increased IL6R signaling caused a 0.107 (95% CI 0.048 to 0.166) increase in the natural log odds of developing CAD. After using TSCMR to adjust for any confounding via Type-2 Diabetes this was attenuated to 0.104 (95% CI 0.053 to 0.156). We therefore found a 3% deflation in the MR estimate of IL6R signaling on CAD risk after accounting for a confounding path via T2D (Table 1). This implies the presence of only very moderate degree of confounding by LD in the effect of IL6R signaling on CAD. Since the change in effect is half the size of the standard error, we believe that our results provide empirical evidence that confounding through a neighbouring T2D signal is not a serious threat to the internal validity of studies exploring the association between IL6R signaling and CAD [15]. There was no evidence of heterogeneity among the three SNPs included in our analysis for either the crude or adjusted outcome datasets (*p* = 0.266 and 0.355, respectively).

## 4. Discussion 

Here, we have proposed a novel sensitivity analysis, Two-Sep *cis*-MR (TSCMR), for attenuating bias in *cis*-MR settings in the absence of many independent genetic variants. Our simulations validated this application of two-step methods for *cis*-MR by showing that bias was essentially completely attenuated in the setting where there was a single known biasing pathway. Our additional simulation implies that TSCMR can also be extended to adjust for bias in settings where there are multiple known independent biases. In our applied example, we found, contrary to expectation, that there was not strong evidence of a confounding effect of T2D on the association between IL6R signaling and CAD.

TSCMR differs from Two Step MR because while the latter uses two MR estimates to calculate the indirect effect of an exposure on an outcome, TSCMR uses one MR estimate and two variant-phenotype estimates to estimate the direct effect of the variant on the outcome. Likewise, unlike Two Step MR, and any other MR sensitivity analyses we are aware of, TSCMR adjusts the GWAS summary statistics used in the MR analysis, rather than directly correcting the MR effect estimate. Finally, and most importantly, we have here shown that TSCMR can be used to attenuate bias in settings where there are insufficient independent causal variants to implement existing methods, like multivariable-MR. 

### Limiations and Assumtions of TSCMR

As with any MR method, TSCMR assumes that the assumptions of instrumental variables are valid (i.e., that the variants are strongly associated with the exposure, only causes the outcome through the exposure or confounder, and that there is no unmeasured confounder of the variant-outcome association). Researcher’s interested in point estimation will additionally have to make the monotonicity or No Simultaneous Heterogeneity assumptions [16,17], both of which are untestable in a summary data setting. If two-sample MR is used, then we additionally require that all samples are drawn from comparable populations [18]. 

Since TSCMR involves the implementation of two MR analyses (one to estimate the confounder-outcome association, and one to estimate the exposure-outcome association), the validity of the adjusted effect estimate will depend on the validity of both MR analyses. Hence, in our applied example, we used weighted median to estimate the confounder-outcome association to address any concern that that estimate might be biased by pleiotropy. As with multivariable MR, TSCMR requires that we have correctly specified the pathway(s) in which the bias occurs.

The product and difference of coefficients methods used by TSCMR also make strong assumptions, including: (1) No residual confounding in any estimates. This is likely to be valid in an MR setting because of the random inheritance of genetic variants. (2) None of the confounders of the mediator-outcome association are influenced by the exposure. This is also likely because, by estimating the mediator-outcome association using MR, the major confounder would be residual population structure. Covariate adjustment is also less likely to introduce collider bias than in traditional applications because GWAS typically do not adjust for heritable phenotypes. (3) No regression dilution bias due to non-differential measurement error. However, MR is robust to non-differential measurement error in the exposure [19]. 

The product and difference of coefficients methods can produce unreliable estimates when the mediator or outcome are not continuous. In such an instance we suggest following the existing convention in the MR literature for binary exposures, where the variant-phenotype associations are interpreted as the (continuously varying linear) effect of the variant on the lifetime liability for the phenotype [20]. Where necessary, simple transformations to the GWAS effect estimates, have been suggested to aid interpretability [20]. Thus, since both the mediator and outcome in our applied example are binary, the correct interpretation is that we estimated the associations of genetically predicted IL6R signaling on CAD, after adjusting for the liability to T2D.

The application of both the product and difference method and Two-Sample MR in this context will additionally assume linearity and no effect modification for all effect estimates used. While both assumptions are untestable in a summary data context, MR estimates will provide a valid estimate of the average causal effect even in the presence of non-linearities [21]. Finally, both the bootstrapped SE and the propagation of error SE we calculate will be biased if the true standard error is not normally distributed.

## 5. Conclusions and Future Directions

Here, we have introduced a novel method, two-step *cis*-MR (TSCMR), for attenuating bias in a *cis*-MR setting. Unlike existing methods like multivariable-MR, our method can successfully attenuate bias even in the absence of multiple independent genetic variants to use as instruments. This will therefore allow researchers to correct for biases in *cis*-MR settings where they could previously only test for its presence using colocalization. To facilitate the implementation of Two-Step *cis*-MR, we have developed the TwoStepCisMR R package (available from https://github.com/bar-woolf/TwoStepCisMR/wiki, acessed on the 25 August 2022).

In our implied example, we found empirical evidence that T2D is not a serious confounder of the association between IL6R signaling and CAD. This provides important support for proposals to use IL6R as a therapeutic target for CAD, and we would therefore suggest further research to confirm this proposal. 

We hope that Two-step *cis*-MR will be a useful sensitivity analyses for MR researchers.

## Figures and Tables

**Figure 1 genes-13-01541-f001:**
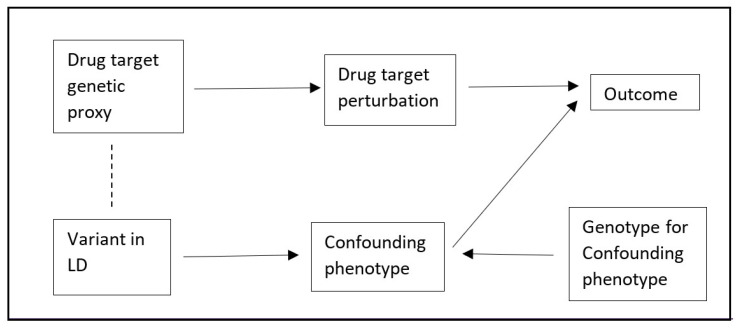
Directed Acyclic Graph (DAG) representing confounding by linkage disequilibrium in a *cis*-Mendelian randomisation analysis.

**Figure 2 genes-13-01541-f002:**
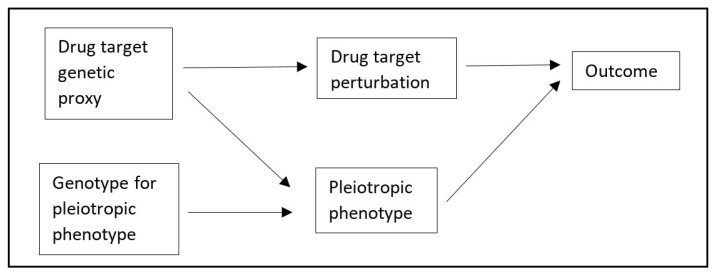
Directed Acyclic Graph (DAG) representing bias through pleiotropy in a *cis*-Mendelian randomisation analysis.

**Table 1 genes-13-01541-t001:** Results of simulation and applied example.

	Crude Estimate	TSCMR Estimate	% Difference	Crude Estimate SE	TSCMR PE SE	TSCMR BS SE
Simulation *	1.377	1.000	27%	0.142	0.175	0.175
IL6R & CAD	0.107	0.104	3%	0.030	0.026	0.026

Crude = MR estimates using the unadjusted variant-outcome association(s), TSCMR = MR estimates after adjusting the variant-outcome association(s) using TSCMR. SE = Standard error. PE = Propagation of Error. BS = Bootstrapped. The % Difference was defined as the % deflation in the TSCMR estimate compared to the crude estimate. * The estimates for the simulation are the mean estimate after 100,000 repetitions. The true casual effect of the exposure on the outcome in the simulation was 1.000, therefore the mean bias in the effects were 0.909 and 0.029 for the crude and TSCMR models, respectively. The Monte-Carlo standard error were <0.001 for all the estimates other than the bias in the crude effect estimate (Monte Carlo SE = 0.004).

**Table 2 genes-13-01541-t002:** Results of simulation to explore the ability of 2SCMR to simultaneously adjust for multiple biasing pathways.

	Bias in Crude Estimate	Bias in TSCMR Estimate	Crude Estimate SE	TSCMR PE SE	TSCMR BS SE
Sim. of Sup. Appendix A	1.336	−0.018	0.174	0.226	0.226
Sim. of Sup. Appendix A	0.406	−0.365	0.174	0.248	0.248

Crude = MR estimates using the unadjusted variant-outcome association(s), TSCMR = MR estimates after adjusting the variant-outcome association(s) using TSCMR. SE = Standard error. PE = Propagation of Error. Appendix A is a setting in which both biasing paths are independent, Appendix A is a setting where the two paths are not independent. All estimates are the average of 100,000 receptions. Monte Carlo standard errors were all < 0.001, with the exception of bias in both crude estimates (0.006 and 0.004 for the simulations of Appendix A, respectively), and the adjusted estimate for the simulation of Appendix A (0.004).

## Data Availability

The data presented in this study are openly available in the OpenGWAS repository at https://gwas.mrcieu.ac.uk/ (accessed on 25 August 2022), reference numbers: ukb-d-30710_raw (CRP), finn-b-T2D (T2D), ebi-a-GCST005195 (CAD) [22]. The R code scripts used in the simulation and applied example are available from https://doi.org/10.17605/OSF.IO/3XG92 (accessed on 25 August 2022). MR analyses in this paper were run using the TwoSampleMR, R package [23,24].

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
