# Peer review of "TwoStepCisMR: A Novel Method and R Package for Attenuating Bias in cis-Mendelian Randomization Analyses"

_genes, 2022, doi:10.3390/genes13091541_

Round 1
Reviewer 1 Report
I have read with interest the original article ,, TwoStepCisMR: A novel method and R package for attenuating bias in cis-Mendelian randomization analyses”. Through simulation experiments and an applied example (the effect of interleukin 6 receptor signaling on coronary artery disease risk), the authors evaluated a novel method and R package for attenuating bias in cis-Mendelian randomization analyses. The results may have important clinical implications and this method can successfully attenuate bias in a cis-MR setting, even in the absence of multiple independent genetic variants to use as instruments.
I consider that the article is well-structured, and the results are presented in an appropriate manner, but I suggest better highlighting the novelty of the study. Also, I suggest including the details of the broader impacts on the study made, addressing the future scope and topics that are important, and also the limitations of the study.
Reviewer 2 Report
In this article, woolf et al. presented a method for attenuating bias in cis-Mendelian randomization analyses. The authors shown the potential of their method that successfully attenuate bias even in the absence of multiple independent genetic variants to use as instruments. Their pilot study found empirical evidence that T2D is not a serious confounder of the association between IL6R and CAD.
The methods section is clearly described and results were supported by the data presented. This manuscript can be an important addition to the literature. The following comments need to be solved.
The authors may provide more information in the introduction section regarding the mendelian randomization analyses with some classical text books as references. This may help the general readers.
The authors may avoid using "simple method" throughout the article.
The limitation of the method should be discussed in detail and a statement indicating this should be added in the abstract.
Looking on https://osf.io/3xg92/ (Institutional repository deposited with codes), I suggest the authors to also deposit some data files that can run the code.
